# The Role of the Inner Nuclear Layer for Perception of Persisting Tiling Inside a Monocular Scotoma

**DOI:** 10.3390/brainsci12111542

**Published:** 2022-11-14

**Authors:** Rishikesh Gandhewar, Neringa Jurkute, Axel Petzold

**Affiliations:** 1Department of Medicine, Imperial College London, South Kensington Campus, London SW7 2AZ, UK; 2Moorfields Eye Hospital, City Road, London EC1V 2PD, UK; 3Queen Square Institute of Neurology, University College London (UCL), London WC1N 3BG, UK

**Keywords:** retinal artery occulusion, tiling, teichopsia, positive scotoma, deformation phosphene, INL, OCT, function structure relationship

## Abstract

We report two patients, one with and one without long-term persistent tiling inside an arcuate macular scotoma. In both cases, the scotoma was caused by a cilioretinal artery occlusion. Both patients were almost identical regarding the location and extent of the scotoma. In both cases, there was a comparable degree of atrophy on optical coherence tomography for the retinal nerve fibre, ganglion cell, and inner plexiform layers. The main difference was the preservation of the inner nuclear layer in the patient with persistent tiling. In this patient, optical coherence angiography demonstrates preserved perfusion of the superior vascular plexus, which was not the case in the patient with the negative scotoma who also had atrophy of the inner nuclear layer. Recreational use of cannabinoid enhanced the intensity of perceived tiling in the relative scotoma of the first patient. A review of the literature suggests that the persistent tiling described in our case is different to teichopsias of retinal or cerebral origin. These data suggest that persistent monocular tiling in a scotoma arises from retinal circuit activity that requires the preservation of the inner nuclear layer. Future research should investigate this functional–structural relationship in other diseases, including glaucoma.

## 1. Introduction

Tiling is a term used to describe a visual pattern on a surface. Here, we describe persisting tiling within the imaginary surface of a visual circumscribed field defect, a relative scotoma. Scotoma are areas of the visual field that are perceived frequently as dark patches or blind spots. Some individuals perceive flashes of light, known as photopsia, a scintillating fortification spectrum or teichopsia. Teichopsias have no definitive cause, but their visual symptoms can be experienced seemingly at random. Early descriptions from Sir J F W Herschel and Dr Hubert Airy describe teichopsia as an angulated coloured or monochrome pattern advancing from the periphery into the central vision, similar to thick running liquid [1]. The temporal evolution of teichopsia in migraine is related to the cytoarchitecture of the visual cortex. The migration of positive and negative visual symptoms through the visual field is caused by cortical spreading depression [2]. Spreading depression was originally described in the retina [3], but in contrast to the cortex, the relationship of visual patterns has not been well described in relation to the retinal cytoarchitecture. One attempt was made by Grüsser, who proposed a retinal feedback loop as the electric origin of binocular deformation phosphenes [4].

Here we present a serendipitous observation of persisting monocular tiling which permits to revisit earlier observations [3,4]. Two patients presented to our service with anatomically almost identical damage to the left retina. The difference in visual perception was that one patient reported a relative scotoma with persisting tiling and the other patient a negative scotoma. The anatomical difference between the two was preservation of the inner nuclear layer (INL) in the first individual and atrophy of the INL in the second. Because tiling persisted over a 7-year clinical follow-up period and anatomical investigations remained stable a relationship between structure and function is likely and will be discussed.

## 2. Materials and Methods

### 2.1. Retinal Imaging

Retinal spectral domain optical coherence tomography (OCT) and OCT angiography (OCTA) were performed using the Heidelberg Spectralis Device (Heidelberg Engineering, Heidelberg, Germany). For longitudinal assessment of the peripapillary retinal nerve fibre layer (pRNFL) thickness, a 3.4 mm ring scan was placed around the optic disc with the eye-tracking function (EBF) enabled for best accuracy as described [5]. The reported intraclass correlation coefficients (ICC) for repeated measurements are 1.00, which is excellent. Likewise, the EBF function was used for repeated measurements of volume scans, which were placed at the optic disc and macular. Automated segmentation was performed using the Heidelberg Spectralis Viewer Software. Pharmacological pupil dilation was not performed. All scans were quality-controlled using the OSCAR-IB criteria [6]. Reporting adhered to the nomenclature of the APOSTEL 2.0 guidelines [7].

### 2.2. Perimetry

The automated 24–2 Humphrey visual fields were recorded using the SITA Standard strategy with a III, white stimulus, and the background set to 31.5 ASB. Fixation was monitored and fixation losses were noted. The false-positive and negative errors were recorded. The Amsler chart was held at a viewing distance of 30 cm. At this distance, one degree of visual angle is accounted for by one square on the Amsler chart.

### 2.3. Pupillometry

The Neuroptics Pupillometer NPi-200 (NeurOptics, Inc., Irvine, CA, USA) was used.

### 2.4. Search Strategy and Selection Criteria

A review of the literature was conducted using Pubmed and Google Scholar. We used the search terms “teichopsia”, “positive scotoma”, “Charles Bonnet Syndrome”, “CBS”, “tessellopsia”, “tiling”, “migrainous aura”, “visual snow”, “cannabis”, “cannabinoid receptors”, and “inner nuclear layer”.

## 3. Results

### 3.1. Case #1

A 42-year-old woman, who presented to an outpatient clinic with a sudden onset, painless scotoma in the left eye (Figure 1A–C). The Humphrey visual field was of good quality with zero fixation losses and an excellent positive and false error rate. There were no peripheral visual field defects. The Amsler chart revealed a semitransparent macular arcuate scotoma in the left eye, extending 4.5 degrees of visual angle horizontally and 1.5 degree vertically (Figure 1B). She was able to see through this relative scotoma, which was not detected by automated perimetry. Within the scotoma, she noticed a pattern of predominately black lines with a faint yellow tint (Figure 1C). There were slow undulations to the line patterns. The pattern of tiling was visible equally against the Amsler chart and against a uniform bright, dark, or differently coloured background. She could not suppress the teichopsia with voluntary horizontal saccades with both eyes closed. Neither did the initiation of the monocular pressure or binocular pressure abolish her teichopsia. She was pressure-blinded during this procedure. Binocular deformation phosphenes could be elicited.

Her past medical history was unremarkable, including no significant history of headache, but she had consumed lysergic acid diethylamide (LSD) and magic mushrooms in her youth. She had not developed any persistent visual symptoms from this hallucinogenic drug exposure.

On examination, unaided high-contrastvisual acuities were 6/4 bilaterally with normal colour vision on Ishihara chart testing. The assessment of the pupil was normal, clinically and by formal pupillometry. The OCT of her good, right eye was normal. Images are shown for the affected left eye in Figure 2. The OCT demonstrated a localised area of loss of the retinal nerve fibre layer (RNFL, Figure 2A), GCL (Figure 2B), and inner plexiform layer (IPL, Figure 2C). There was preservation of the INL (Figure 2D). The extent of atrophy of the inner retinal layers was from the optic disc to the macular in the left eye (Figure 2A–C). The shape of the teichopsia could be superimposed approximately to the area of the slight thickening of the INL (Figure 2C). The OCT assessments were repeated over a 7-year period. The thickness of the INL remained stable (Figure 2E). Likewise, the pRNFL remained stable over the 7 year observation period (Figure 2F).

A transient cilioretinal artery occlusion was thought to be the most likely cause. The retinal structure in the right eye was normal. Blue autofluorescence of the retina was normal in both eyes. The symptoms persisted on long-term follow-up. At follow-up, she reported a new observation: the intensity of the teichopsia could be enhanced by smoking cannabis. An additional OCTA was performed at the age of 49 years, which showed preserved perfusion of the inner retinal vasculature (Figure 3A–C).

### 3.2. Case #2

A 34 year old man with a non-proliferative diabetic retinopathy presented with a scotoma of sudden, painless onset (Figure 4A,B). Initially, there were positive symptoms that disappeared over time. There were no teichopsia. The Humphrey visual field was of good quality without fixation losses, and again there were excellent positive and false-error rates (Figure 4A). There was mild, mainly peripheral sensitivity loss in both eyes. In the left eye, there was a macular arcuate scotoma. The man was not able to see through this scotoma, which appeared as a darkened-out area on the Amsler chart (Figure 4B). The fundus appearance was consistent with a clinical diagnosis of a cilioretinal artery occlusion. He remained under follow-up for monitoring of his diabetic retinopathy.

When reassessed 7 years after onset, he was not aware of the negative scotoma that initially brought him to our attention. On examination, his best corrected high-contrast visual acuities were 6/6 in the right eye and 6/5 in the left eye. Colour vision was normal in both eyes upon Ishihara chart testing. The pupil responses were normal. However, when asked in clinic to initiate monocular and binocular deformation phosphenes, he could again perceive his scotoma. He could elicit this by applying gentle digital pressure to the eye balls or through forceful blinking. The effect was short-lasting. The area of the atrophy of the inner retinal layers is almost identical in location to our first patient, with the sole difference being that atrophy involves the INL (Figure 5A–D). Blue autofluorescence of the retina was normal in both eyes. On OCTA at 7-year follow-up, there was localised vascular drop out of the deep vascular plexus in the area affected by the cilioretinal artery occlusion (Figure 6A–C).

## 4. Discussion

Central nervous system pathology is a common cause of positive scotoma, resulting in binocular symptoms with characteristic presentation patterns [1,2]. It can be challenging to establish whether positive visual phenomena are monocular (retinal) or binocular (cerebral) in origin. In clinical practice, the most frequent presentation is a binocular, migrainous, visual aura. There is typically a fortification spectrum pattern [1,2,8]. Exclusion of central pathology such as stroke is required. Visual snow produces different visual symptoms but is frequently associated with migraine. Visual snow persists for months to years and in some cases is lifelong. The pattern is of flickering dots, likened to the static of an analogue television or falling snow [8]. Finally, there are coloured, formed symmetric patterns following a stroke of the occipital cortex in the blind visual field [9]. They are a variation of visual hallucinations. The most frequent of the visual hallucinations is the Charles Bonnet Syndrome (CBS). Most individuals have good insight into their hallucinations, which typically last for minutes but can persist for longer [10]. A range of images may be perceived in an area of impaired vision, from simple shapes and flashes to nonthreatening complex images, such as faces and animals [10,11]. Whilst CBS is typically binocular, Toosy et al. report a case of monocular CBS in a patient following coronary artery bypass grafting [12]. The patient described seeing young children in solely the left inferior field; however, his hallucinations were relieved when he closed his left eye or averted his gaze. Furthermore, Cogan describes a patient with homonymous hemianopia experiencing zigzag lines alongside formed images following proton-beam irradiation [13]. Overall, whilst the tessellopsia described in CBS may have some similarities with the visual tiling our first patient described, her symptoms were not relieved by eye closure nor by ocular pressure causing deformation scotoma and persisted for 7 years, arguing against CBS as a cause. Similarly, the illusions of patients with Alice in Wonderland syndrome are characterised by metamorphopsia, typically affecting the face, referred to as prosopometamorphopsia [14]. Furthermore, the visual symptoms cannot be explained by previous psychedelic use, but the reported enhancement requires discussion. Hallucinogen-persisting perception disorder causes recurring perception of shapes, flashes, and trail phenomena for up to five years after initial consumption [15]. Primate retinal bipolar cells express the cannabinoid receptors CB1 and CB2 [16]. Experimental, pharmacological dissection changes the electroretinogram of the a- and b-waves in scotopic and photopic conditions. The effect of treatment with the agonists and antagonists of the cannabinoid receptors in humans with persistent teichopsia has not yet been studied. Overall, the tiling pattern described in patient #1 could not be explained by a central cause and could not be suppressed by saccadic eye movements as self-reported for the migrainous visual aura experienced by two neurologists [17]. Whilst this helps to build an argument against the central origin of tiling in our case #1, we acknowledge that there are as yet no cohort studies on the relationship between supposed saccadic suppression and the migrainous visual aura [17]. Ultimately, all visual experience is cortical. This includes the perceived tiled image in patient #1. There are top-down processes (suppression, plasticity, and retinal correspondence) as well as bottom-up processes (scotoma, vision loss, and ocular misalignment) occurring in visual perception. Yet, in both our patients the primary cause is most likely retinal.

Expanding on the discussed spectrum, these positive visual symptoms are different to the typical negative scotomata experienced with lesions to the retina in our second patient. Rather, the monocular presentation seen with our patient is most likely explained by retinal pathology, which can cause tiling as a positive phenomenon in the relative scotoma. This is supported by the structural data. The localised atrophy of the inner retinal layers on OCT matches the area affected by the relative, arcuate macular scotoma. Retinal migraine results in transient monocular visual impairment lasting from minutes to hours. Widespread variations in teichopsia have been reported, ranging from simple zigzag lightening patterns to complex patterns described as “black paint dripping” [18]. The pattern of retinitis pigmentosa is of an annular photopsia, developing in the early years of life [19]. Various causes of autoimmune retinopathy can lead to positive scotoma, though they are always rapidly progressive and generally develop bilaterally [20]. Acute zonal occult outer retinopathy (AZOOR) occurs most commonly in young adult females of Caucasian origin, causing photopsia but not teichopsia [21]. The fundus autofluorescence in AZOOR is striking and was normal in our cases. Hence, although the pathology can be localised to the retina, the scotoma characteristics in patient #1 differed from the typical presentations of retinal disease.

One limitation of our case study is that we did not perform tangent perimetry or microperimetry of the scotoma. Likewise, we are unable to comment in more detail on the residual vision within the, strictly speaking, relative scotoma of case #1. Another shortcoming of our report is that we have not conclusively investigated perceptual filling-in [22]. It would also be interesting to investigate how stimuli of the surrounding intact visual field affect what is perceived within the scotoma using scanning-laser ophthalmoscopy for the determination of the absolute fovea position [23]. Finally, we did not make use of the multifocal electroretinogram investigation of the arcuate macular scotoma, neither did we investigate retinal function during light and dark adaptation as was previously done for eyeball deformation activity [4]. Such investigations would require ethical approval for a research study over and above what is possible within routine clinical care.

The data in our first case suggest that the incomplete atrophy of the IPL and thickened INL on OCT permitted the survival of a small proportion of horizontal, amacrine, and bipolar cells. Spontaneous electrical activity may feed retrograde into the preserved bipolar cells of the thickened INL and from there laterally into the surrounding area, thereby exciting the ganglion cells. This prevents the development of a negative scotoma as seen with the more complete cilioretinal artery occlusion, causing dense atrophy, including the INL, as seen in our second patient.

In conclusion, the preservation of the INL probably maintains a feedback loop of electrical activity that is perceived as a described pattern of persistent tiling. Retinal spreading depression [3] in the relative scotoma may be the cause for the persistent undulating pattern of visual tiling. Our observation may be of interest to other ophthalmological conditions where macular arcuate scotoma with the preservation of the INL occur, such as glaucoma.

## Figures and Tables

**Figure 1 brainsci-12-01542-f001:**
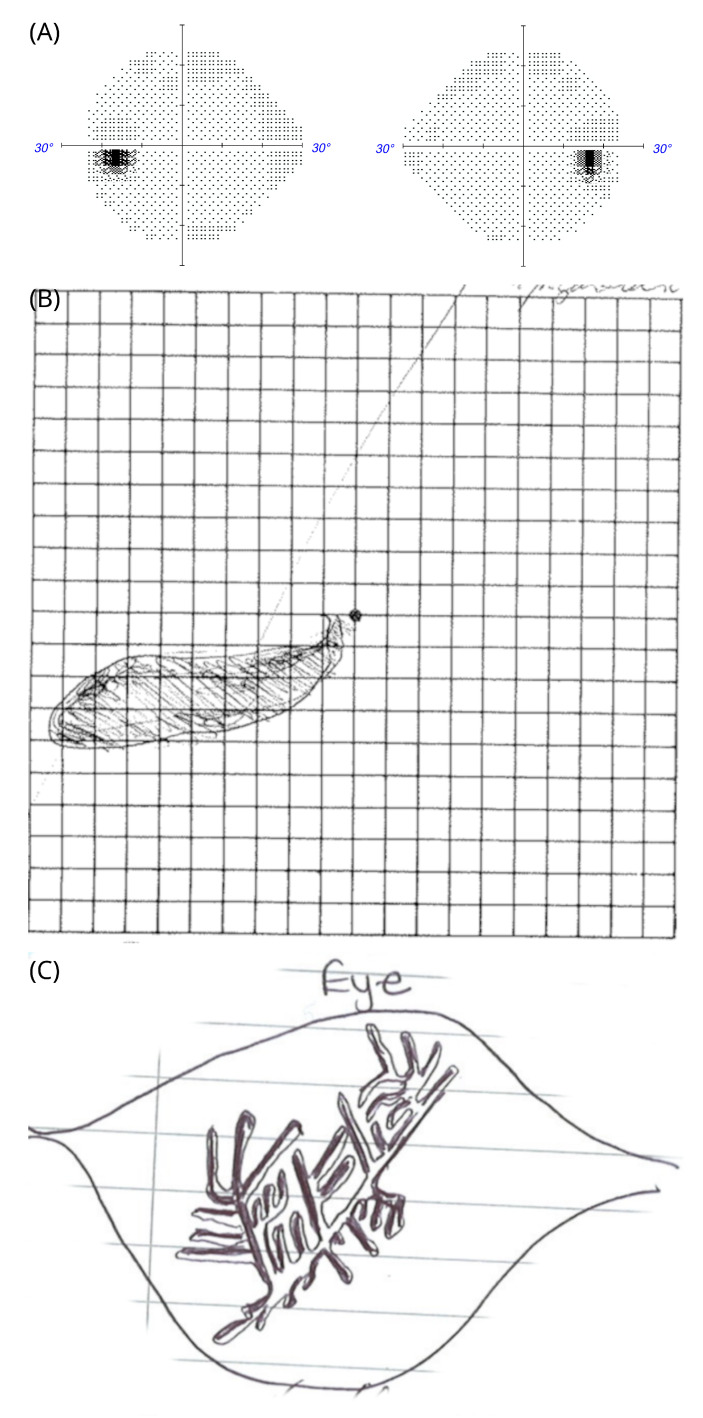
Tiling inside a relative scotoma in patient #1. (**A**) Automated perimetry showing the Humphrey SITA standard protocol with a white size III stimulus. (**B**) relative macular arcuate scotoma on the Amsler chart. She can see the grid lines through the scotoma. (**C**) Tiling pattern as sketched by the patient within the scotoma of her left eye on a white piece of paper with thin greyish lines. She described the tiling pattern as constantly moving lines. The colour of the lines was described “yellowish” against different backgrounds.

**Figure 2 brainsci-12-01542-f002:**
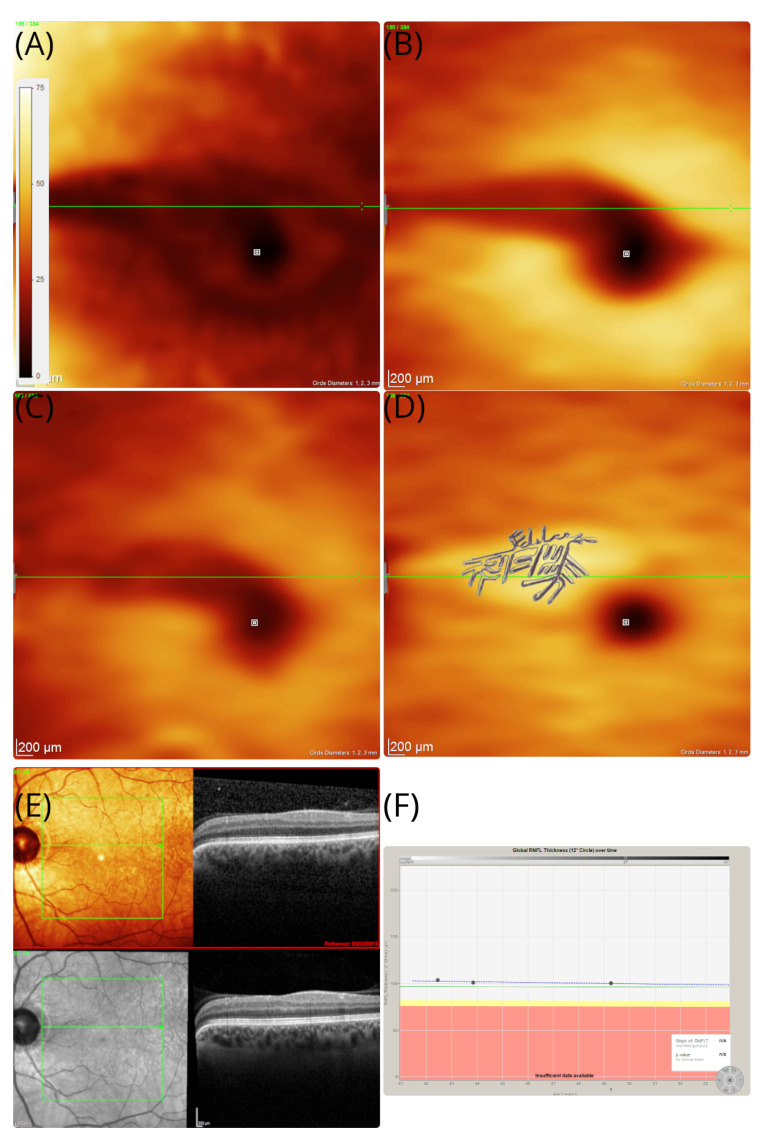
OCT in patient #1. The OCT thickness maps of the left eye revealed an area of sectorial atrophy of the retinal nerve fibre (**A**) RNFL (dark shaded arcuate area in the macula), (**B**) GCL (dark shaded), and (**C**) IPL (dark shaded). However, the (**D**) inner nuclear layers remains preserved and even slightly thickened (brighter yellow arcuate area in the macular). The area of atrophy stretches from the temporal optic disc margin to the macular, which corresponds with the macular arcuate scotoma on the Amsler chart in Figure 1B. The tiling pattern perceived by the patient is superimposed to the INL. The colour coding of all thickness maps ranges from black (=minimal cellular thickness, 0 μm) to white (=maximal cellular thickness, 75 μm). (**E**) In addition to the volume scan thickness maps presented here, B-scans are shown from baseline (red frame) and 7-year follow-up. The horizontal green line is placed at the same location in subfigures (**A**–**E**). (**F**) Except for the sectorial atrophy of the RNFL in subfigure (**A**), there is no evidence for pRNFL atrophy elsewhere, which remained stable across the 7-year observation period.

**Figure 3 brainsci-12-01542-f003:**
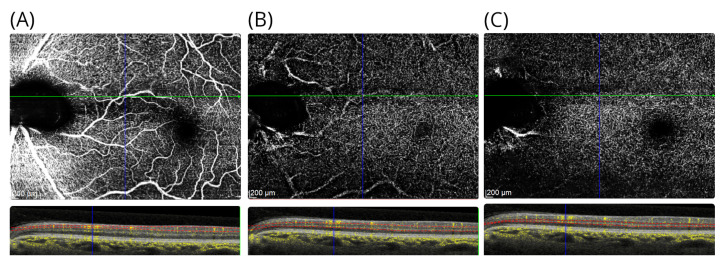
OCTA in patient #1. The OCTA shows the (**A**) the inner retinal vasculature at level of the RNFL, (**B**) the superficial vascular plexus above the INL, and (**C**) the deep vascular plexus located below the INL. For each OCTA, the layer segmentation (red dashed lines) and OCTA signal (yellow) are shown. Note that there are shadowing artefacts from the larger inner retinal vessels in subfigures (**B**,**C**).

**Figure 4 brainsci-12-01542-f004:**
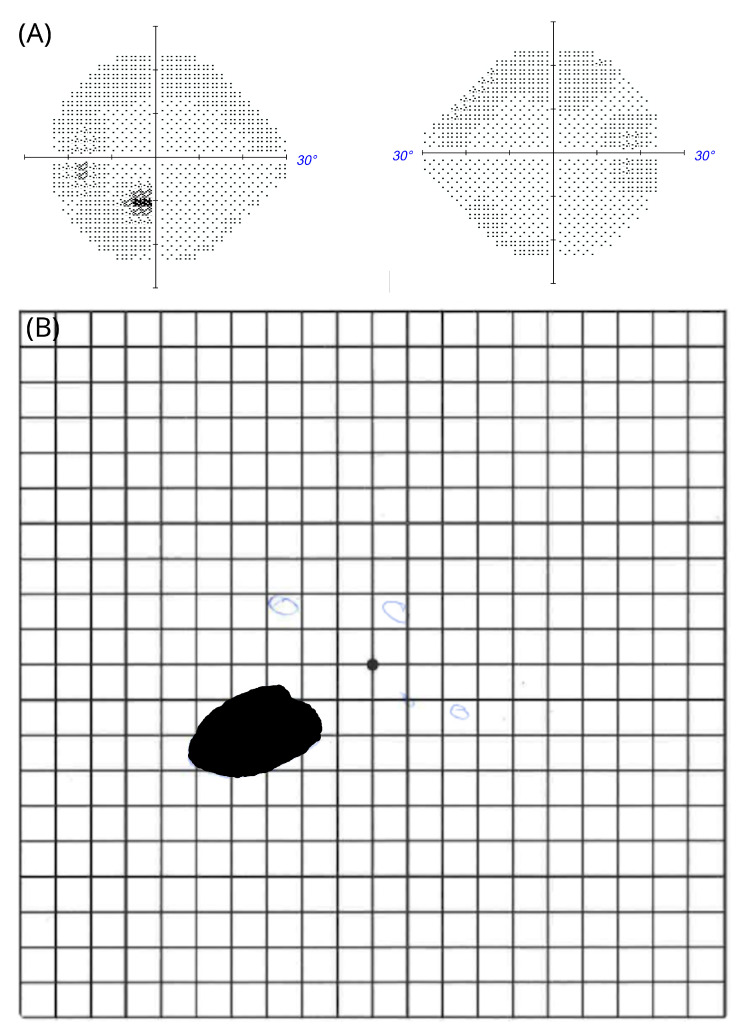
Negative scotoma in patient #2 (**A**) Automated perimetry; (**B**) macular arcuate scotoma on the Amsler chart.

**Figure 5 brainsci-12-01542-f005:**
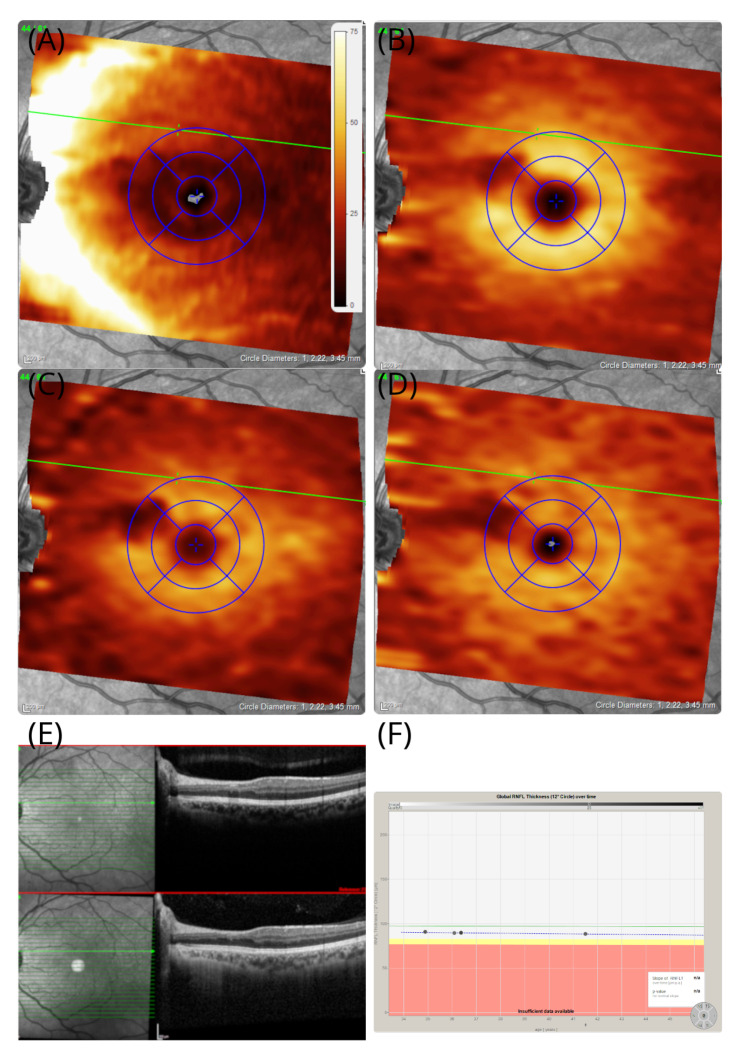
OCT in patient #2 The OCT images of the left eye revealed that an area of localised atrophy of the retinal nerve fibres (**A**) RNFL (**B**) GCL, (**C**) IPL, and (**D**) INL remains preserved. The area of atrophy stretches from the temporal optic disc margin to the macular, which corresponds with the macular arcuate scotoma on the Amsler chart in Figure 4B. (**E**) In addition to the volume-scan thickness maps presented here, B-scans are shown from baseline (red frame) and 7-year follow-up. (**F**) Except for the sectorial atrophy of the RNFL in subfigure (**A**), there is no evidence for pRNFL atrophy elsewhere, which remained stable across the 7-year observation period.

**Figure 6 brainsci-12-01542-f006:**
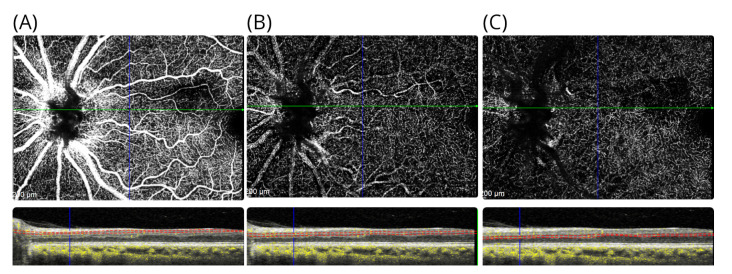
OCTA in patient #2. The OCTA shows the (**A**) the inner retinal vasculature at level of the RNFL, (**B**) the superficial vascular plexus above the INL, and (**C**) the deep vascular plexus located below the INL. For each OCTA, the layer segmentation (red dashed lines) and OCTA signal (yellow) are shown.

## Data Availability

Not applicable.

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
