# Peer review of "The Role of the Inner Nuclear Layer for Perception of Persisting Tiling Inside a Monocular Scotoma"

_brainsci, 2022, doi:10.3390/brainsci12111542_

Round 1

Reviewer 1 Report

I enjoyed this thoughtful exploration of cases that raises a new hypothesis.  The authors found two cases of localised inner retinal thinning (presumed ischaemia), very similar in every way except one had persisting linear patterns within their visual scotoma, associated with a thicker inner nuclear layer (INL).  This correspondence between structure and function is a compelling observation and I think it warrants reporting, but these two cases only provide a limited amount of evidence.  It is clearly an intelligent discussion, but I think the authors have overly discounted the possibility that the linear "tiling" image is cortical in origin.

Perhaps your journal specifies the structure of the manuscript and the listing of methods and figures, but the flow of the paper is affected by the technical listings.  I think it would be nicer to have an introduction, list the imaging and functional testing devices, then describe the cases without these extra sections.

I think there might be too many figures. I suppose it is an editorial decision but perhaps the normal data from the fellow eye could be omitted, so figure 1 would show the greyscale and total deviation from the affected eye, the Amsler grid and patient illustration.  Figure 2 could show parts D, E and F.  In Figure 3 part A can be omitted, and all the blue and green location indicators should be in the same place, the center of the lesion, for all angiograms.  A cross section along the affected slice could be shown to demonstrate that the OCTA segmentation correctly follows the superficial and deeper retinal vascular plexi.  Figure 4 could likewise show the greyscale and total deviation of the affected eye, and the Amsler grid.  Figure 5 could show only B, C, D.  This would simplify the potential confusion that case 2 has no nerve fibre defect, which some readers could see as as another structural difference between the cases to explain the difference in symptoms (although not a credible hypothesis).  Figure 6 could just show B, C, D with a cross section to verify the segmentation, and the blue/green location markers should all be centered on the lesion.  Figure 7 should be removed, the figure is not required to prove stability.  The sections shown, indicating the layers of thickening and thinning relative to the unaffected retina, should be part of Figures 2 and 5.  I think Figure 8 is excessive too, and could be removed.

With these changes I think section 3.3 and 3.4 could be removed.

In the discussion, I think it would help to state how difficult it can be to establish whether positive visual phenomena are monocular (retinal) or binocular (cerebral), simply because they are present in the dark or with eyes closed.  Here there is obvious correlation between the retinal pathology and the visual symptom, and so clearly the primary cause of the scotoma is retinal, and yet the source of the tiled linear image could yet be cortical.  Some aspects, like the lack of change with lighting, background or indeed the deformation of the closed eye, would seem to argue that the image is cortical.  I see that there are differences between her symptom and the previous reported monocular CBS, and that it did not disappear with saccades under closed eyelids, but these arguments are not definitive.  The paragraph on other retinal causes of positive visual phenomena highlights how this tiling image is more similar to cortical diseases than retinal pathology.

This observation of a preserved INL, indeed a thickened INL, in the eye with the tiling experience is interesting.  I think the discussion takes the issue of deformation phosphenes too far, as ocular deformation did not affect the symptom for this patient, so it is not especially relevant.  Rather than arguing about horizontal elements in the retina, perhaps it is sufficient to say that this preserved and possibly thickened INL, containing bipolar, horizontal and amacrine cells from within the affected area, would appear to be contributing to the development of this tiling image experience.  This could be a persistent derangement of the inputs to surviving ganglion cells, or something unusual about the nature of the surviving ganglion cells, or the altered receptive fields of the ganglion cells outside the lesion.

There are top down processes (suppression, plasticity, retinal correspondence) as well as bottom up processes (scotoma, vision loss, ocular misalignment) occurring in visual perception.

Author Response

I enjoyed this thoughtful exploration of cases that raises a new hypothesis.  The authors found two cases of localised inner retinal thinning (presumed ischaemia), very similar in every way except one had persisting linear patterns within their visual scotoma, associated with a thicker inner nuclear layer (INL).  This correspondence between structure and function is a compelling observation and I think it warrants reporting, but these two cases only provide a limited amount of evidence.  It is clearly an intelligent discussion, but I think the authors have overly discounted the possibility that the linear "tiling" image is cortical in origin.

Perhaps your journal specifies the structure of the manuscript and the listing of methods and figures, but the flow of the paper is affected by the technical listings.  I think it would be nicer to have an introduction, list the imaging and functional testing devices, then describe the cases without these extra sections.

A: We want to thank the referee for the insightful comments. Please find below our point-to-point reply starting with A:...

I think there might be too many figures. I suppose it is an editorial decision but perhaps the normal data from the fellow eye could be omitted, so figure 1 would show the greyscale and total deviation from the affected eye, the Amsler grid and patient illustration. 

A: Thank for for your insightful advice on simplifying the figure.

Figure 2 could show parts D, E and F. 

A: Thank you for the suggestions and also your points on Figure 7. We have now changed and merged both figures.

In Figure 3 part A can be omitted, and all the blue and green location indicators should be in the same place, the center of the lesion, for all angiograms. A cross section along the affected slice could be shown to demonstrate that the OCTA segmentation correctly follows the superficial and deeper retinal vascular plexi. 

A: part A has been omitted. All blue and green location indicators are now in the same place. A cross-section of the segmentation is shown in the new subfigure (D).

Figure 4 could likewise show the greyscale and total deviation of the affected eye, and the Amsler grid. 

A: The figure as has been revised as suggested.

Figure 5 could show only B, C, D.  This would simplify the potential confusion that case 2 has no nerve fibre defect, which some readers could see as as another structural difference between the cases to explain the difference in symptoms (although not a credible hypothesis). 

A: Thank you, together with your point on Figure 7 and your point above on Figure 2, these images have been simplified and are presented in a comparable layout to the new composite figure 2.

Figure 6 could just show B, C, D with a cross section to verify the segmentation, and the blue/green location markers should all be centered on the lesion. 

Figure 7 should be removed, the figure is not required to prove stability.  The sections shown, indicating the layers of thickening and thinning relative to the unaffected retina, should be part of Figures 2 and 5. 

A; Thanks for the suggestion. Figure 7 has been removed and the sub figures integrated to Figures 2 and 5.

I think Figure 8 is excessive too, and could be removed.

A: Figure 8 has been removed.

With these changes I think section 3.3 and 3.4 could be removed.

A: Sections 3.3. and 3.4 have been removed as suggested.

In the discussion, I think it would help to state how difficult it can be to establish whether positive visual phenomena are monocular (retinal) or binocular (cerebral), simply because they are present in the dark or with eyes closed. 

A: Excellent suggestion. We have added this point as “It can be challenging to establish whether positive visual phenomena are monocular (retinal) or binocular (cerebral) in origin.” as the second sentence in the discussion.

Here there is obvious correlation between the retinal pathology and the visual symptom, and so clearly the primary cause of the scotoma is retinal, and yet the source of the tiled linear image could yet be cortical. 

A: The referee is correct. Ultimately all visual experience will be cortical. We have clarified this in the discussion as: “Ultimately, all visual experience is cortical. This includes that the perceived tiled image in patient \#1. There are top down processes (suppression, plasticity, retinal correspondence) as well as bottom up processes (scotoma, vision loss, ocular misalignment) occurring in visual perception. Yet, in both our patients the primary cause is most likely retinal.”

Some aspects, like the lack of change with lighting, background or indeed the deformation of the closed eye, would seem to argue that the image is cortical. 

A: We agree with the referee that this might potentially imply change in gain, but we do not know how this could be tested convincingly in these two individuals. We propose to keep this point open with the added sentence from your previous point ot our discussion.

I see that there are differences between her symptom and the previous reported monocular CBS, and that it did not disappear with saccades under closed eyelids, but these arguments are not definitive.  The paragraph on other retinal causes of positive visual phenomena highlights how this tiling image is more similar to cortical diseases than retinal pathology.

A: “We agree, none of the psychophysic tests performed can be considered as definite. They all are subjective. Therefore the strong point of the paper is that the OCT remained structurally identical over a 7 year observation period.”

This observation of a preserved INL, indeed a thickened INL, in the eye with the tiling experience is interesting.  I think the discussion takes the issue of deformation phosphenes too far, as ocular deformation did not affect the symptom for this patient, so it is not especially relevant. 

A: The referee also recommended to delete Figure 8 which is relevant to support the deformation phosphenes argument. We have deleted both the part of the discussion and the figure.

Rather than arguing about horizontal elements in the retina, perhaps it is sufficient to say that this preserved and possibly thickened INL, containing bipolar, horizontal and amacrine cells from within the affected area, would appear to be contributing to the development of this tiling image experience. 

A: Thank you for guidance on shortening and increasing eloquence. This has be phrased in the discussion as “ The data in our first case suggest that the incomplete atrophy of the IPL and thickened INL on OCT permitted for survival of a small proportion of horizontal, amacrine and bipolar cells. Spontaneous electrical activity may feed retrograde into the preserved bipolar cells of the thickened INL and from there laterally into the surrounding area, thereby exciting the ganglion cells.”

This could be a persistent derangement of the inputs to surviving ganglion cells, or something unusual about the nature of the surviving ganglion cells, or the altered receptive fields of the ganglion cells outside the lesion.

There are top down processes (suppression, plasticity, retinal correspondence) as well as bottom up processes (scotoma, vision loss, ocular misalignment) occurring in visual perception.

A: Thank you for this excellent summary which we have also included into our discussion as already cited in one of your previous points.

Reviewer 2 Report

This is a report of two cases of cilioretinal artery occlusion presenting with scotoma. The aim was to describe the relationship between visual function and retinal structure. The observation and interpretation provided by the authors are interesting, although larger cohort are needed to validate those findings.

-It would be interesting to report the evolution of ILM thickness from diagnosis to last follow-up visit, and maybe represent it in a plot comparing the two cases.

-Please add fundus photographs for case 1 and 2 at the time of diagnosis of cilioretinal artery occlusion.

-Line 28: Please replace “thought” with “through”.

-Line 89: Add “lysergic acid diethylamide” before (LSD).

-Line 99: Delete “from”.

-Line 176: replace “him” by “In”.

Author Response

This is a report of two cases of cilioretinal artery occlusion presenting with scotoma. The aim was to describe the relationship between visual function and retinal structure. The observation and interpretation provided by the authors are interesting, although larger cohort are needed to validate those findings.

A: We thank the referee for the careful reading of our report and the suggestions to improve. Please find below our point-to-point reply, starting with A:...

-It would be interesting to report the evolution of ILM thickness from diagnosis to last follow-up visit, and maybe represent it in a plot comparing the two cases.

A: The ILM remained unchanged over time. With referee #1 trying to reduce the number of figures we have not compiled a new composite figure for the ILM.

-Please add fundus photographs for case 1 and 2 at the time of diagnosis of cilioretinal artery occlusion.
A: We do not have fundus photographs from the time of diagnosis of cilioretinal artery occlusion.

-Line 28: Please replace “thought” with “through”.

A: done.

-Line 89: Add “lysergic acid diethylamide” before (LSD).

A; done.